# Genome-Wide Identification and Analysis of the *WRKY* Gene Family in *Asparagus officinalis*

**DOI:** 10.3390/genes14091704

**Published:** 2023-08-27

**Authors:** Jing Chen, Sijia Hou, Qianqian Zhang, Jianqiao Meng, Yingying Zhang, Junhong Du, Cong Wang, Dan Liang, Yunqian Guo

**Affiliations:** 1College of Biological Science and Technology, Center for Computational Biology, National Engineering Laboratory for Tree Breeding, Beijing Forestry University, Beijing 100083, China; cjkn23731@163.com (J.C.); hsj381552790@163.com (S.H.); mjq990521@163.com (J.M.); zhangyingying@bjfu.edu.cn (Y.Z.); dujunhong0324@163.com (J.D.); w2286687005@163.com (C.W.); 2Chinese Institute for Brain Research, Beijing 102206, China; awayzqq@163.com; 3College of Biological Sciences, China Agricultural University, Beijing 100193, China; 4The Tree and Ornamental Plant Breeding and Biotechnology Laboratory of National Forestry and Grassland Administration, Beijing Forestry University, Beijing 100083, China

**Keywords:** *Asparagus officinalis*, *WRKY* family, genome-wide analysis, chromosome localization, synteny analysis

## Abstract

In recent years, the related research of the *WRKY* gene family has been gradually promoted, which is mainly reflected in the aspects of environmental stress and hormone response. However, to make the study of the *WRKY* gene family more complete, we also need to focus on the whole-genome analysis and identification of the family. In previous studies, the whole *WRKY* gene family of *Arabidopsis*, legumes and other plants has been thoroughly studied. However, since the publication of *Asparagus officinalis* genome-wide data, there has never been an analysis of the whole *WRKY* gene family. To understand more broadly the function of the *WRKY* gene family, the whole genome and salt stress transcriptome data of asparagus were used for comprehensive analysis in this study, including *WRKY* gene family identification, phylogenetic tree construction, analysis of conserved mods and gene domains, extraction of cis-acting elements, intron/exon analysis, species collinearity analysis, and *WRKY* expression analysis under salt stress. The results showed that a total of 70 genes were selected and randomly distributed on 10 chromosomes and one undefined chromosome. According to the functional classification of *Arabidopsis thaliana*, the *WRKY* family of asparagus was divided into 11 subgroups (C1–C9, U1, U2). It is worth considering that the distribution rules of gene-conserved motifs, gene domains and introns/exons in the same subfamily are similar, which suggests that genes in the same subfamily may regulate similar physiological processes. In this study, 11 cis-acting elements of *WRKY* family were selected, among which auxin, gibberellin, abscisic acid, salicylic acid and other hormone-regulated induction elements were involved. In addition, environmental stress (such as drought stress and low-temperature response) also accounted for a large proportion. Interestingly, we analyzed a total of two tandem duplicate genes and 13 segmental duplication genes, suggesting that this is related to the amplification of the *WRKY* gene family. Transcriptome data analysis showed that *WRKY* family genes could regulate plant growth and development by up-regulating and down-regulating gene expression under salt stress. Volcanic maps showed that 3 and 15 *AoWRKY* genes were significantly up-regulated or down-regulated in NI&NI+S and AMF&AMF+S, respectively. These results provide a new way to analyze the evolution and function of the *WRKY* gene family, and can provide a reference for the production and research of asparagus.

## 1. Introduction

Transcription factors (TFs), proteins that regulate downstream genes by binding to specific DNA, play an important role in plant growth and development [1].

As one of the largest transcription factor families, the functions of the *WRKY* family in environmental stress and hormone response have been thoroughly studied [2]. WRKY TFs are divided into three groups based on the number of WRKY domains and the difference in zinc finger structure: Group 1 contains two WRKY domains and one C_2_H_2_ zinc finger motif. Unlike group 1, group 2 lacks one WRKY domain. Group 3, like group 2, contains a WRKY domain and a zinc finger motif; however, this zinc finger structure is C_2_HC. It is also worth noting that group 2 is subdivided into five subfamilies [3]. Both WRKY amino acid domains and zinc finger motifs contain DNA-binding domains that can specifically recognize W-box (TTGACT/C) sequences. The binding of the WRKY domain to the DNA domain after mutation was significantly reduced. Interestingly, WRKY proteins can also recognize domains such as PRE4 (TGCGCTT), WT-box (GGACTTTC), and WK-box (TTTTCCAC) [2,3,4,5].

Due to the frequent exposure to various biological and abiotic stresses, such as salt stress, drought stress, pests and diseases, plants have evolved molecular mechanisms to prevent and resist stress, among which *WRKY* family is an important part of the resistance to environmental stress. A WRKY transcription factor named SPF1 (Sweet potato factor 1) was discovered in Ipomoea batatas more than 20 years ago [6]. Subsequently, ABF1 (ABRE binding factor 1) and ABF2 (ABRE binding factor 2) involved in seed germination were found in *Avena sativ* [7]. After extensive research, the researchers identified the *WRKY* family in several species, including *A.thaliana* (74 *WRKY* genes), *Oryza sativa* (102 *WRKY* genes), and *Solanum lycopersicum* (81 *WRKY* genes) [8,9,10]. Under drought stress, *AtWRKY53* in *A. thaliana* controls stomatal opening and closing by binding Qua-Quine Starch promoter, thus affecting drought tolerance [11]. The overexpression of *WRKY11* in rice can enhance plant heat resistance [12]. The *WRKY* family mainly regulates plant stress processes. However, different studies have found that the *WRKY* family is also involved in regulating pollen development, and fruit ripening [13,14]. The study of FUSCA3 in *A*. *thaliana* showed that AtWRKY TFs can activate FUSCA3 by regulating W-box, thus participating in seed dormancy process [15]. *AtWRKY44* (*TRANSPARENT TESTA GLABRA2*) is the first and so far only WRKY TF that can induce trichoid development [16,17]. Interestingly, in the vernalization pathway of plants, *AtWRKY* can promote the accumulation of CULLIN3A promoter to activate the expression of flowering genes. However, *WRKY* can also inhibit flowering through the non-vernalization pathway in the latest investigation, indicating that *WRKY* has a dual role in the induction of flowering [18,19]. In summary, the WRKY transcription factor family regulates the plant life process from multiple dimensions. As a perennial herb, asparagus is cultivated all over the world, especially in Western developed countries. As the largest producer and exporter of asparagus, China’s asparagus planting area accounts for more than 2/5 of the world, which makes asparagus also have a certain economic value [20]. *A. officinalis* is best known for its medicinal properties in addition to being a food ingredient. Flavonoids, as the main components of asparagus, have anti-inflammatory, antibacterial and antioxidant effects. The leading anti-inflammatory ingredient in flavonoids is rutin, and it has also been suggested to relieve diabetes and arthritis, as well as liver disease, which is also worth mentioning [21,22,23,24,25]. The categories of asparagus can be easily classified according to color, such as green asparagus, white asparagus, etc. [26,27]. According to the type of asparagus, the global production of asparagus in 2021 reached 8.5 million metric tons [28]. The asparagus genome was published in 2017 and includes 10 chromosomes with a total length of 1187.54 Mb [29]. Salinity is an important factor affecting the physical and chemical properties of land, and its imbalance will aggravate the soil environmental stress in arid areas, resulting in crop yield reduction [30]. Quantitative analysis results show that the global saline-alkali land area is more than 424,291.05 square kilometers [31]. Therefore, it is urgent to increase the efforts to cultivate salt-resistant crops. Previous studies have shown that asparagus has a strong salt tolerance and can survive in saline soil with a concentration below 0.3% [20]. Previous studies focused on the edible and medicinal value of asparagus and rarely involved genome-wide analysis. Since the genomic information has been published, we combined the genomic information to identify the WRKY transcription factor family in order to provide a basis for the study of asparagus and WRKY in other plants. In the following studies, we conducted a comprehensive bioinformation analysis of the asparagus *WRKY* family, including the identification and screening of gene families, comparison of phylogenetic relationships with *A. thaliana*, analysis of gene structure and conserved motif, analysis of cis-acting elements, collinearity analysis within and between species, and *WRKY* expression under salt stress. The purpose and significance of this study is to deepen the understanding of WRKY family and explore the regulatory network of *WRKY* gene in asparagus, so as to provide a certain basis for salt stress resistance and the improvement of asparagus yield. 

## 2. Materials and Methods

### 2.1. Identifification of the WRKY Gene Family in A. offificinalis

The *A. offificinalis* genome sequence file, annotation file, protein sequence and coding sequence (CDS) used in this paper were obtained from the National Center for Biotechnology Information (NCBI) (https://www.ncbi.nlm.nih.gov/, accessed on 20 August 2022). The Hidden Markov Model (HMM) file of the WRKY domain (PF03106) was obtained from the Pfam database (release 35.0; http://pfam.xfam.org/, accessed on 20 August 2022) [32]. The candidate members of *A. offificinalis* with the WRKY protein domain were screened using the HMMMER search (Version 3.2.1) method in two steps. First, the HMM profile in the HMMER software is used to search for and compare target members that contain the WRKY protein domain (Appendix A). And in Linux, MAFFT v7.505 was used for multiple sequence alignment of candidates. Second, to avoid missing the target protein, candidate proteins with e-value < 1 × 10^−20^ in the first screening were selected to reconstruct the HMM model using HMMER (version 3.2.1) [33]. The new HMM model was used to search the protein sequence with an e-value < 0.05. We considered the intersection of the two results as the final filter result. Further, we verified the existence of the WRKY domain in candidate proteins on the online website of the SMART program (http://smart.embl-heidelberg.de/, accessed on 23 August 2022), NCBI Conserved Domain Search (https://www.ncbi.nlm.nih.gov/Structure/cdd/wrpsb.cgi, accessed on 23 August 2022), and Pfam Batch Sequence Search (http://pfam.xfam.org/search#tabview=tab1, accessed on 23 August 2022), and determined the final protein sequence after screening and removing redundancy [34].

### 2.2. Phylogenetic Analysis and Classification of AoWRKY Genes

To explore the phylogenetic relationships and taxonomy of *WRKY* Genes, we established a rooted neighbor-joining (NJ) phylogenetic tree between *A. offificinalis* (*AoWRKY*) and *A. thaliana* (*AtWRKY*) in MEGA11 software (version 11.0.11). The AtWRKY protein sequences from TAIR (https://www.Arabidopsis.org, accessed on 25 August 2022) were reported by Eulgem [2]. The *AoWRKY* family members and *AtWRKY* family members are aligned under the initial parameters using ClustalW in MEGA11 software. The neighbor-joining (NJ) method was adopted and 1000 replications were used for the bootstrap method, Poisson model, and pairwise deletion [35,36]. Finally, The phylogenetic tree of *AoWRKY* genes was glorified with online software iTOL (http://itol.embl.de/, accessed on 11 September 2022) and Adobe Illustrator 2020 software (version 24.0.1.341). 

### 2.3. Gene Structure, Cis-Acting Regulatory and Conserved Motif Analysis of AoWRKY Genes

The online programs MEME (https://meme-suite.org/meme/tools/meme, accessed on 12 September 2022) and PlantCARE (http://bioinformatics.psb.ugent.be/webtools/plantcare/html/, accessed on 12 September 2022) were used to analyze the conserved motif and cis-acting regulatory of the *AoWRKY* family, respectively. The following parameters were set for conservative motif analysis: Maximum Number of Motifs 15, Motif E-value Threshold no limit, Minimum Motif Width 6, Maximum Motif Width 50, Minimum Sites per Motif 2, and Maximum Sites per Motif 70. At the same time, the structural information of *AoWRKY* genes was extracted from GFF files, and the above results were visualized in the same image with TBtools software [3].

### 2.4. Chromosome Localization, Duplication and Synteny Analysis

The chromosome length and location information of the *WRKY* genes were obtained from NCBI. TBtools (v1.0987671) software was used to predict and locate the location of *AoWRKY* genes. Theoretical isoelectric point (pI), molecular weight (MW), amino acid numbers (aa) instability index, and aliphatic index were analyzed in ProtParam in Expasy (https://web.expasy.org/protparam/, accessed on 3 September 2022) and DNAMAN software (version 9.0). The open reading frame (ORF) lengths were searched in the ORFfinder website (https://www.ncbi.nlm.nih.gov/orffifinder, accessed on 13 September 2022) and the subcellular localization of the *AoWRKY* genes was confirmed in the BUSCA program (https://busca.biocomp.unibo.it, accessed on 13 September 2022) [37,38].

### 2.5. Analysis of AoWRKY Gene Expression Profiles under Different Conditions

*AoWRKY* expression levels under different salt stress conditions were experimentally treated as follows: (1) non-inoculated *A. officinalis* plants without salinity stress (NI); (2) inoculated *A. officinalis* plants without salinity stress (arbuscular mycorrhiza fungi, AMF); (3) non-inoculated *A. officinalis* plants subjected to salinity stress (NI + S); and (4) inoculated *A. officinalis* plants subjected to salinity stress (AMF + S) [39]. Sequence read archives (SRAs) were retrieved from the National Centre for Biotechnology Information (NCBI) (https://www.ncbi.nlm.nih.gov/sra/?term=SRP188664#, accessed on 14 September 2022). FASTQ files generated the pair-end data containing forward and reverse reads from SRA files. FastQC and MultiQC are used for data inspection [40,41]. We filtered low-quality fragments and process adapters through Trimmomatic [42]. We constructed a genome index file using Hisat2 and compared reads to the reference genome [43]. The DEseq2 package in the R language is used to screen differentially expressed genes [44]. Screening of differentially expressed genes was carried out at a significant adjusted *p*-value (*p*.adj) < 0.05 and an absolute value of log2FC (log of fold change) > 1 to filter out insignificantly expressed genes. The volcano plot was created by using differential expression data from ggplot. Differentially expressed *AoWRKY* genes were labeled in the volcano plot. The heatmap of the differentially expressed genes was constructed by the pheatmap package.

## 3. Results

### 3.1. Identification of AoWRKY Genes in A. offificinalis

After two filters by HMMMER search (Version 3.2.1), we obtained a total of 96 target genes. These 96 candidate genes were submitted to the online site of the SMART program, NCBI Conserved Domain Search, and Pfam Batch Sequence Search for the identification of WRKY domains. A total 70 final target genes were identified through domain identification and the removal of repeats. Chromosomal mapping of the target gene was performed using TBtools, which named the gene *AoWRKY1-70* based on its location on the chromosome. The basic physical and chemical properties of target genes are shown in Table 1. *AoWRKY52* had the smallest amino acid number (aa) and molecular weight (114aa, 12,996.32 Da). In contrast, the protein sequence with the largest amino acid number and molecular weight (MW) was *AoWRKY32* (674aa and 72,655.57 Da). In addition, the theoretical isoelectric point (pI) and MW of *AoWRKY4* and *AoWRKY34* could not be predicted, and the pI of the remaining *AoWRKY* genes ranged from 4.66 (*AoWRKY29*) to 10.24 (*AoWRKY38*). Among all the protein sequences, 29 were acidic (pI < 6.5), 33 were alkaline (pI > 7.5), and six were neutral (pI = 6.5–7.5). Instability Index values above 40 are labelled as structural instability. In the *AoWRKY* sequence, only *AoWRKY11* belongs to the stable category, and the others are all unstable. Subcellular localization prediction of the *AoWRKY* family revealed that only *AoWRKY44* was located in the chloroplast, 47 were located in the nucleus, and 22 in extracellular space.

### 3.2. Systematic Classification Analysis of AoWRKY and AtWRKY

*A*. *thaliana* has been well studied as a model plant. Therefore, the *Arabidopsis* WRKY family was selected as an outgroup to construct a new evolutionary tree. To analyze the classification and function of the *AoWRKY* gene family (Appendix A), MEGA11 software was used to perform multiple sequence alignments between 75 *AtWRKY* genes of *Arabidopsis* and 70 *AoWRKY* genes. In addition, the results were visualized by constructing a rooted neighbor-joining phylogenetic tree (Figure 1). The evolutionary tree shows that members classified into 7 subfamilies (I, II-a, II-b, II-c, II-d, II-e, III) in *Arabidopsis* have been more finely divided into 11 subfamilies(C1–C9, U1, U2) in *Asparagus* [2]. Subgroups I and II-c are subdivided into subgroups C1 & C9, C4 & C7. However, the U1–U2 subgroups included only the members of *A. offificinalis* (*AoWRKY17*, *AoWRKY62*, *AoWRKY63*). This suggested that these three genes may have mutated or evolved. It is worth noting that genes grouped into the same subfamily may regulate the same life process. For instance, *AtWRKY14*, *AtWRKY16*, *AtWRKY35* and *AtWRKY69* in C5 subgroups were involved in the pression of plant thermomorphogenesis, while *AtWRKY28*, *AtWRKY50*, *AtWRKY51* and *AtWRKY71* in C4 subgroups were responsible for inducing the biosynthesis of different hormones(TAIR, https://www.arabidopsis.org/, accessed on 15 September 2022). Interestingly, although the number of members contained in each subfamily was different (the number of subfamily members ranged from 1 to 32), the ratio of *AoWRKY* to *AtWRKY* in each subfamily was almost 1:1, indicating that the evolution of *WRKY* gene families in the two plants was relatively conservative, and the genetic relationship was relatively close.

### 3.3. Chromosome Localization and Cis-Acting Regulatory Analysis of AoWRKY

The *AoWRKY* family was named *AoWRKY1-70* based on chromosome length information and *AoWRKY* location information from the genome annotation file of *A. offificinalis*, while the results were visualized in TBtools (Figure 2). The figure shows 70 *AoWRKY* genes randomly distributed across 10 chromosomes and one undefined chromosome. We subsequently found that chromosomes 1, 2, 5, 7, and 8 contain most members of the *AoWRKY* family (84.3%), whereas only one gene is located on chromosomes 9, 10, and Un.

There is a variety of cis-acting elements around 2000 bp upstream of the transcription start site, which regulate gene expression mainly by binding to transcription factors. The *AoWRKY* family 2000 bp promoter sequence was submitted in the PlantCARE online program, and the results are shown in Appendix A. The prediction results of cis-acting elements after screening showed that 11 cis-acting elements were distributed in 70 *AoWRKY* genes, including 294 the MeJA (Methyl Jasmonate) responsiveness elements (distributed in 57 *AoWRKY*), 60 salicylic acid responsiveness elements (distributed in 41 *AoWRKY*), 129 anaerobic induction elements (distributed in 53 *AoWRKY*), 232 abscisic acid responsiveness elements (distributed in 63 *AoWRKY*), 29 defense and stress responsiveness elements (distributed in 24 *AoWRKY*), 51 gibberellin-responsive elements (distributed in 30 *AoWRKY*), 41 low-temperature responsiveness elements (distributed in 26 *AoWRKY*), 124 light-responsive elements (distributed in 57 *AoWRKY*), 25 auxin-responsive elements (distributed in 20 *AoWRKY*), 54 *MYB*-binding sites involved in drought inducibility (distributed in 41 *AoWRKY*), and six wound-responsive elements (distributed in 6 *AoWRKY*).It can be concluded that the *AoWRKY* family mainly regulated plant hormone response and abiotic stress, so as to provided reference for the functional study of the *AoWRKY* family. 

### 3.4. Gene Structure and Conserved Motif Analysis of AoWRKY Genes

In order to explore the function of the *AoWRKY* gene family in detail, we submitted the data of the evolutionary lineage, conserved motif, gene domains, and intron/exon structure of the *AoWRKY* family to the Gene Structure View of TBtools (Figure 3). We found that the evolutionary relationship that differed from Figure 1 was two genes in the C9 group in Figure 3A (*AoWRKY8*, *AoWRKY52*), which is not classified in the same subgroup, but in a separate group. This phenomenon indicates that although the two genes have high homology with *AtWRKY19* in C9, they have a relatively low homology with each other.

The number of conserved motifs in group C1 was the highest (8), while the number of conserved motifs in group C9 was the lowest (2). Analysis of conserved motifs showed that *AoWRKY* genes in the same subfamily contained almost the same conserved motifs, which meant that they participated in similar regulatory processes. Motif1 and motif2 are found in almost all *AoWRKY* families, while motif15 is only found in the C5 subfamily (Figure 3B).

Seven conserved domains exist in the *AoWRKY* family, and WRKY domains exist in the entire *AoWRKY* family (Figure 3C). C5, C7, C8, U1, and U2 only have WRKY domains, but C6 also contains plant_zn_clust domains. It is worth noting that *AoWRKY59* is in C1 and *AoWRKY29* is in C4. The *AoWRKY61* in C3 has only the WRKY superfamily domain.

Figure 3D shows that each gene of *AoWRKY* family contains more introns, especially the C1, C2, C3, and C8 subfamilies, which provides a powerful entry point for us to better understand the gene structure.

### 3.5. Tandem Gene Duplication and Segmental Gene Duplication of AoWRKY Genes

Gene duplication includes whole-genome duplication, segmental duplication and tandem duplication [3]. We imported *A. offificinalis* genome data and the genome annotation file into the TBtools plugin MCScanX, which enabled us to obtain collinear information and gene duplication information of the *A. offificinalis* genome(Figure 4). Two tandem duplication gene pairs were derived from the *AoWRKY* family, namely *AoWRKY47* and *AoWRKY48* in the C4 subfamily from Chromosome 7, and *AoWRKY64* and *AoWRKY65* in the C8 subfamily from chromosome 8. To more intuitively display the evolutionary information of two pairs of tandem duplication genes, we calculated their Ka/Ks values (Table 2). We think that when Ka/Ks < 1, the gene evolution process has a negative selection effect, which means that the gene is subject to purification selection. In the segmental gene duplication phase, 1547 pairs were derived, of which only 13 pairs of segmental duplication genes located in Chr1, Chr3, Chr4, Chr5, Chr7, Chr8, and Chr10 were predicted by the *AoWRKY* family.

Not only did genetic evolution occur within *A. offificinalis* species, we also analyzed the homologous gene pairs of *A. offificinalis* with *A*. *thaliala* (11 orthologous gene pairs), *Populus trichocarpa* (58 orthologous gene pairs), *Sesamum indicum* (38 orthologous gene pairs), and *Ananas comosus* (31 orthologous gene pairs) in an attempt to understand the evolutionary relationships between species (Figure 5). The homologous gene pairs between *A. offificinalis* and the other four species are shown in the Supplementary Appendix A. Remarkably, there are four homologous gene pairs shared between the five species. In addition, while there were 14 pairs of homologous gene pairs in *A. offificinalis*, *Sesamum indicum* (*S. indicum*), *Ananas comosus* (*A.comosus*) and *Populus trichocarpa* (*P.trichocarpa*), there were only two pairs in *A. offificinalis*, *S. indicum* and *A. comosus* (Figure 6).

### 3.6. Analysis of AoWRKY Gene Expression under Different Salt Stress

The heat map (Figure 7A,B) and the volcano plot (Figure 7C,D) were drawn in R language (R-4.3.1) editor Rstudio based on the existing RNA-seq data (NCBI). In the experiment with salinity as the only variable, there were 46 differentially expressed genes in the NI and NI + S groups. Obviously, the expression levels of 14 *AoWRKY* genes (*AoWRKY4*, *AoWRKY7*, *AoWRKY11*, *AoWRK12*, *AoWRKY15*, *AoWRKY32*, *AoWRKY33*, *AoWRKY34*, *AoWRK35*, *AoWRKY36*, *AoWRKY40*, *AoWRKY44*, *AoWRKY49*, and *AoWRKY70*) were higher under salt stress (NI + S). In the NI group, the expression levels of eight *AoWRKY* genes (*AoWRKY1*, *AoWRK3*, *AoWRKY6*, *AoWRKY13*, *AoWRKY14*, *AoWRKY25*, *AoWRKY27*, and *AoWRKY30*) were higher. Moreover, we found that the highly expressed genes in the two groups were completely different; so, we concluded that the mechanisms and functions of *AoWRKY* family in response to salt stress may be quite different. Compared with the salt-stress-only group (NI + S), *AoWRKY11/12* in the group (NI) was significantly down-regulated (*p*.adj < 0.05, log2FC > 1), while *AoWRKY30* was significantly up-regulated (*p*.adj < 0.05, log2FC < −1), which was consistent with the expression pattern in Figure 7A,C [45]. Similar to the analysis of NI and NI+S groups, we found that 11 *AoWRKY* genes (*AoWRKY1*, *AoWRKY9*, *AoWRKY13*, *AoWRKY14*, *AoWRKY17*, *AoWRKY27*, *AoWRKY30*, *AoWRKY41*, *AoWRK51*, *AoWRKY64*, *AoWRKY69*) were highly expressed in the AMF group, while 12 genes (*AoWRKY2*, *AoWRKY11*, *AoWRKY15*, *AoWRKY16*, *AoWRKY18*, *AoWRKY21*, *AoWRKY31*, *AoWRKY43*, *AoWRK46*, *AoWRKY49*, *AoWRKY50*, and *AoWRKY68*) were highly expressed in the AMF + S group, and the high-expression genes in the two groups were also not overlapping. In addition, compared with the AMF + S group, six genes in the AMF group were significantly up-regulated and nine genes were significantly down-regulated (Figure 7D).

## 4. Discussion

Asparagus, a lily family, has high edible and medicinal value, and is widely cultivated throughout the globe. Asparagus has some tolerance to several major environmental stresses. However, few previous studies have studied the physiological mechanism of asparagus from the whole-genome level. Based on the published whole-genome information and transcriptome information, this paper conducted a comprehensive analysis of the asparagus WRKY family, aiming to provide references for the cultivation and yield increase in asparagus [46].

As one of the largest transcription factor families, the *WRKY* gene family is mainly involved in plant stress response and hormone regulation [2]. With the rapid development of whole-gene sequencing technology, the *WRKY* family of a variety of plants has been identified and analyzed, including soybean [5], *Solanum lycopersicum* [47], *Carica papaya* [48], cotton [49], *Cucumis sativus* [50], *Brachypodium distachyon* [51], *Gossypium raimondii* [52], *Aegilops tauschii* [53], *Gossypium* [54], *Camellia sinensis* [55], and grape [56], but at present, there is little characterization and analysis of the *WRKY* gene family of *A. offificinalis*.

To understand more broadly the function of the *WRKY* gene family, the whole genome and salt-stress transcriptome data of *A. offificinalis* were used for comprehensive analysis in this study, including *WRKY* gene family identification, phylogenetic tree construction, analysis of conserved motifs and gene domains, extraction of cis-acting elements, intron/exon analysis, species collinearity analysis, and *WRKY* expression analysis under salt stress.

The most critical step in the study was to screen and identify *WRKY* family members from the whole *A. offificinalis* genome for subsequent analysis. Unlike previous studies, there were only 70 *AtWRKY* genes in *A. offificinalis*, compared to 105 in *cabbage* [32] and 74 in *Arabidopsis* [2]. In higher plants, the number of *WRKY* family genes is at least 22, but at most 202, indicating that the evolution of the *WRKY* family belonging to the middle number of *A. offificinalis* has a certain conservation, but also has a high gene-loss rate (http://plntfdb.bio.uni-potsdam.de/v3.0/fam_mem.php?family_id=WRKY, accessed on 13 September 2022). It is worth mentioning that the number of *WRKY* gene family members is not proportional to the size of the genome [57,58]. To further understand the function of *AtWRKY*, we localized the *AtWRKY* family to 10 chromosomes and one undefined chromosome and mapped 11 subfamily (C1–C9, U1, U2) phylogenetic trees of *A. offificinalis* and *Arabidopsis* by multi-sequence alignment. Except for the U1 and U2 subgroups, the number of members of Arabidopsis and asparagus in subgroups C1–C9 is almost 1:1. This suggests that the rates of gene replication and loss in the two species are similar and conserved. Notably, we found no family members of *Arabidopsis* in the U1 and U2 subfamilies, suggesting that our asparagus WRKY family may have mutated and evolved. Because of the structural diversity, the distribution of introns and exons can reflect the phylogenetic relationship of gene families [59]. The number of introns in the *AtWRKY* gene ranges from 2 to 6, and the intron/exon structure is similar in the same subfamily. Similarly, the conserved motifs of WRKY proteins in the same subfamily are similar.

It was found that the *AoWRKY* gene family has a variety of hormone-related regulatory elements, of which more than 57 members have methyl jasmonate and abscisic acid response elements. Methyl jasmonate can induce the expression of defense-related genes in plants, and promote the production of defense substances to resist environmental stresses such as salt stress and low temperature stress. At the same time, it can also conduct signal transduction and promote the synthesis of other hormones [60]. Abscisic acid is also a key factor in plants’ regulation of stress and can enhance plant drought tolerance [61]. In addition, elements involved in drought response were detected in 41 *AoWRKY* genes, and defense and stress response elements were detected in 24 *AoWRKY* genes.

The most critical driving force of species evolution is gene replication, including segment duplication and tandem duplication [62]. A total of two pairs of tandem duplication and 13 pairs of segment duplication were detected in the *A. offificinalis WRKY* family. Not surprisingly, two of the genes in each pair were from the same subfamily. It is more strongly suggested that the *WRKY* gene is more conserved in the same subfamily. 

Arbuscular mycorrhizal fungi have been identified as soil amendments to increase plant salt tolerance due to salt stress [63]. After grouping asparagus and measuring its transcriptome data (Appendix A), we found that *AoWRKY11* and *AoWRKY15* were up-regulated in NI+S group and AMF+S group, indicating that they were salt-stress-related genes. The genes with an up-regulated expression in the NI group were almost contained in the AMF group. In addition, the AMF group also up-regulated the expression of other genes, suggesting that AMF promotion could promote the expression of more genes. Most of the up-regulated genes of the NI + S and AMF + S groups were different, suggesting that AMF could improve gene salt tolerance and enhance gene expression. Unfortunately, the three biological duplications of some gene expression levels were not perfect, and the reproducibility of the data was not ideal. We speculated that these genes might be caused by some mistakes in the experiment, so these genes were not discussed in this study.

## 5. Conclusions

In this work, 70 *AoWRKY* genes were identified, randomly distributed across 11 chromosomes. Further analysis showed that the gene conserved motifs, gene structure and intron/exon arrangement were similar in the same subfamily, indicating highly conserved genes.

We found that the *AoWRKY* family has more hormone response elements and stress defense elements, which indicates that the *AoWRKY* family is mainly involved in hormone regulation and stress resistance. In addition, we also found that this part responds to light-response elements and low-temperature-induced response elements, suggesting that we have functional diversity in the *AoWRKY* family. Finally, we found that several genes were up-regulated or down-regulated under salt stress, and the most important one was that *AoWRKY11/15* was finally identified as a salt-stress-resistant gene.

## Figures and Tables

**Figure 1 genes-14-01704-f001:**
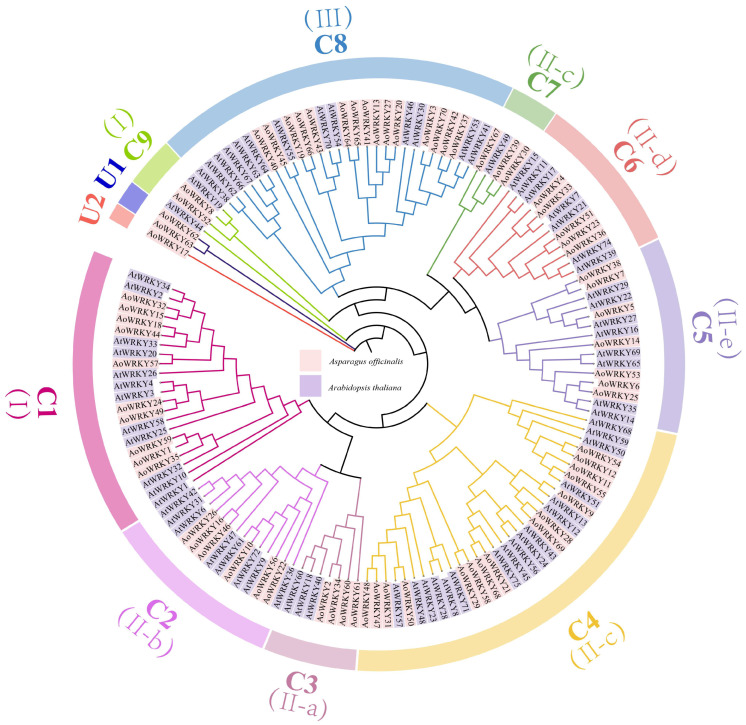
Phylogenetic tree of *WRKY* genes in *A. officinalis* and *A. thaliana*. The neighbor-joining (NJ) method was adopted and 1000 replications for bootstrap method, Poisson model, and pairwise deletion. The *AoWRKY* and *AtWRKY* gene families were labeled with pink and purple blocks, respectively. The evolutionary tree is divided into 11 subfamilies (C1–C9 fa, U2). The Roman numerals (I, II(a,b,c,d,e), III) in parentheses indicate the subfamily classification in *Arabidopsis*, as well as the different domains of the WRKY family.

**Figure 2 genes-14-01704-f002:**
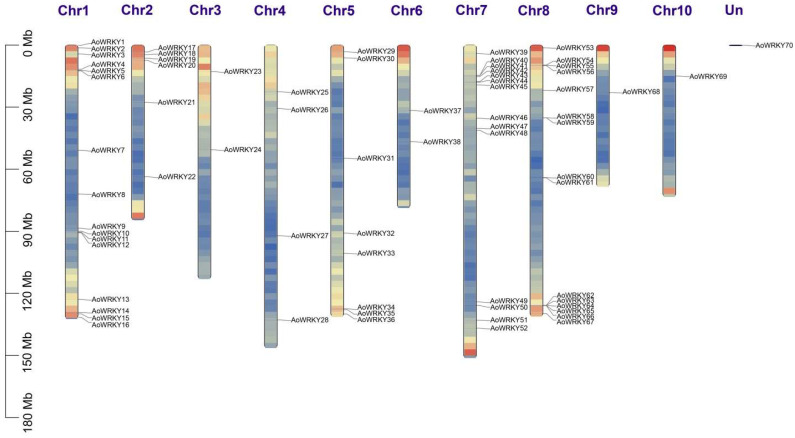
Location distribution of *WRKY* gene family in chromosomes in *A. officinalis*. The ordinate is chromosome length; Chr1-10 and Un stand for chromosome number.

**Figure 3 genes-14-01704-f003:**
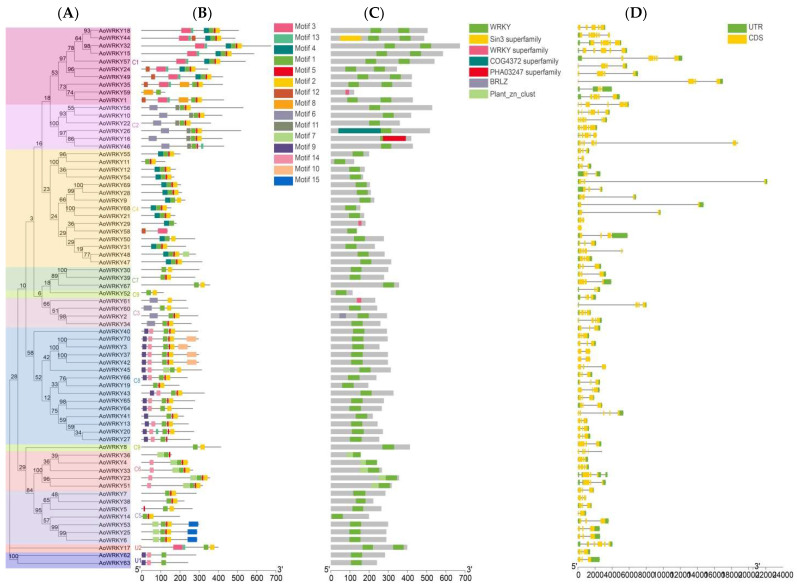
Phylogenetic relationship, conserved motifs, gene structure and domain architecture of *AoWRKY* genes. (**A**) An unrooted NJ tree obtained using MEGA 11 based on *A. offificinalis* WRKY protein sequences. (**B**) Conserved motifs of *AoWRKY* gene family analyzed in MEME. (**C**) Conserved domains of *AoWRKY* family. (**D**) Introns and exons of the *AoWRKY* family. Note: Different colors indicate different meanings, see the figure note for details.

**Figure 4 genes-14-01704-f004:**
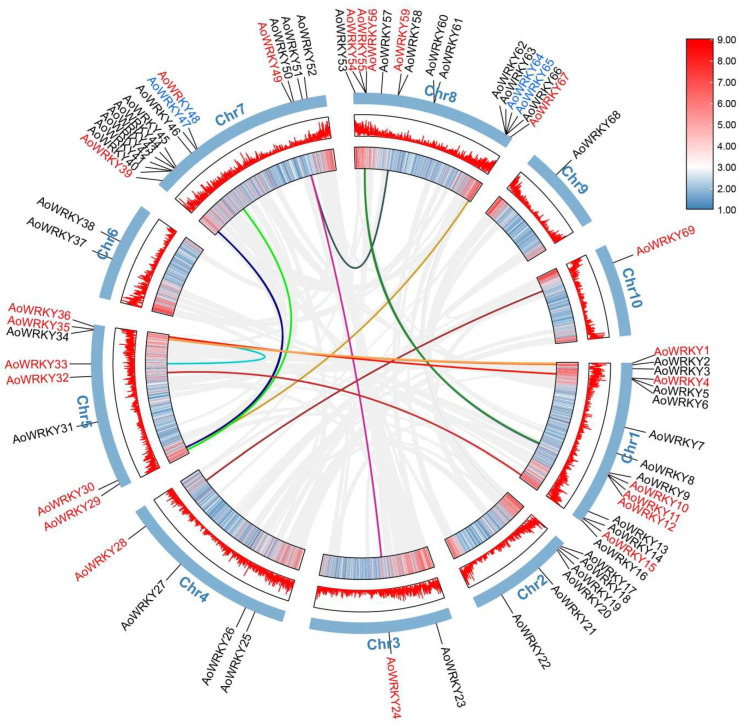
Duplication and synteny of *AoWRKY* genes. The gray lines represent tandem duplicate pairs of whole genome. The colored lines represent segmental gene duplication. The red font indicates segmental gene duplication and the blue font indicates tandem gene duplication. The segmental duplication gene pairs are shown in Appendix A.

**Figure 5 genes-14-01704-f005:**
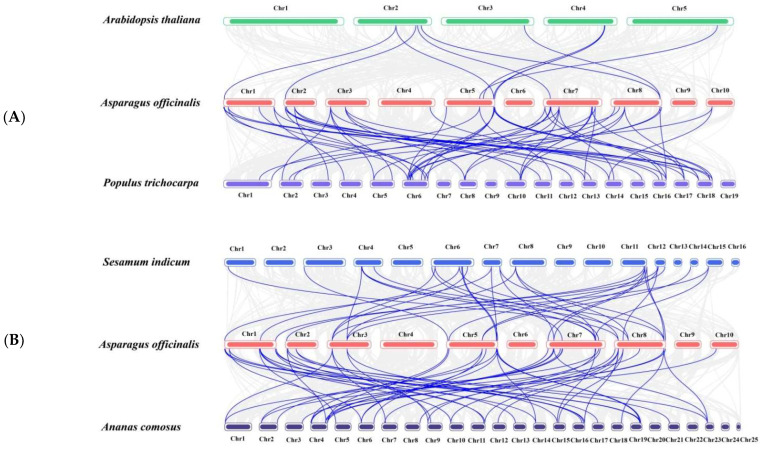
Schematic diagram of syntenic analysis of *AoWRKY* genes. Synteny of the *AoWRKY* genes with the *WRKY* genes of *A. thaliana* & *P. trichocarpa* (**A**), *S. indicum* & *A. comosus* (**B**). The gray lines represent genome-wide collinear gene pairs, and the blue lines represent collinear gene pairs in the WRKY gene family.

**Figure 6 genes-14-01704-f006:**
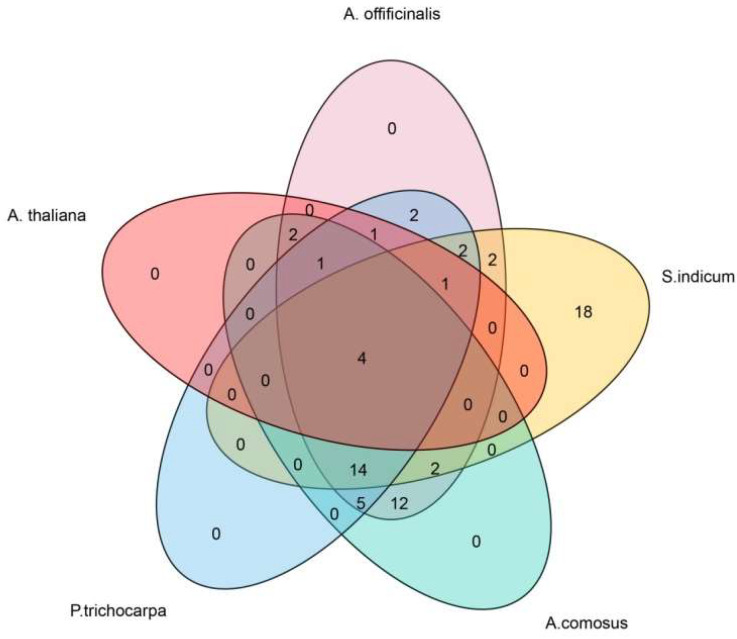
Venn diagram analysis between *A. offificinalis*, *S. indicum*, *A. comosus* and *P. trichocarpa*. The number of overlapping areas of species indicates the number of homologous genes between these species.

**Figure 7 genes-14-01704-f007:**
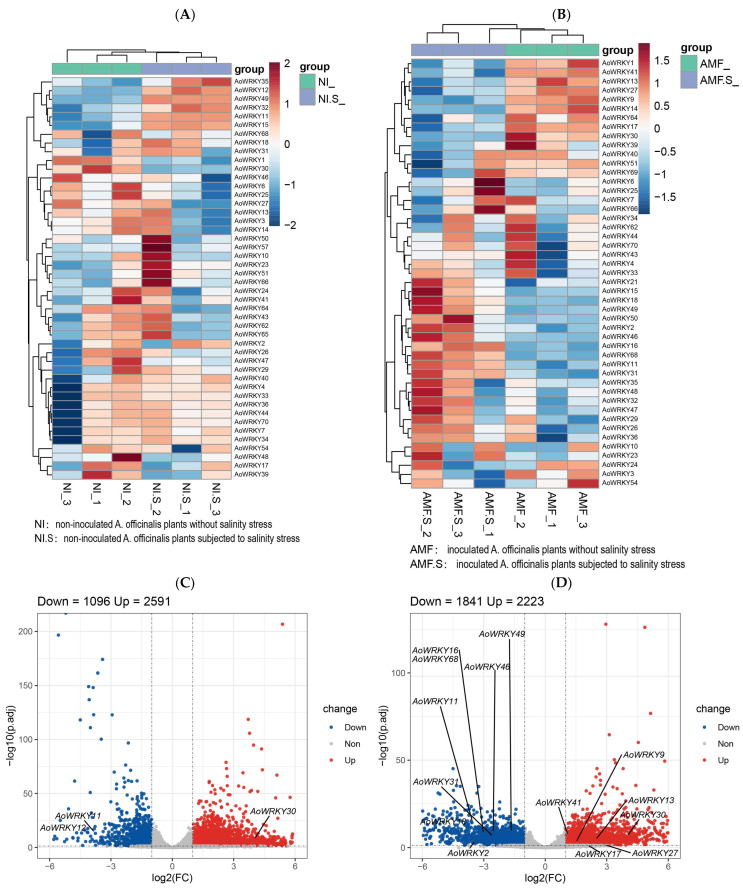
RNA-seq analysis of *AoWRKY* gene family under different salt stress. (**A**,**B**) Expression heatmap of *AoWRKY* gene family under different salt stress. (**C**,**D**) Volcanic plot of *AoWRKY* gene family under different salt stress. The vertical dashed lines represent the points 1 and −1. The horizontal dashed lines represent the points 0.05. Down indicates that the gene is down-regulated and Up indicates that the gene is up-regulated.

**Table 1 genes-14-01704-t001:** Detailed information of *AoWRKY* genes.

Gene Symbol	pI	MW (Da)	Length (aa)	Instability Index	Aliphatic Index	Subcellular Localization	ORF
*AoWRKY1*	6.48	46,154.82	428	55.77	53.18	Nucleus	1287
*AoWRKY2*	9.15	32,384.89	293	59.03	62.32	Nucleus	882
*AoWRKY3*	8.75	28,313.14	254	44.03	64.53	Extracellular	765
*AoWRKY4*	undefined	undefined	241	52.57	65.19	Nucleus	726
*AoWRKY5*	5.24	29,765.87	264	46.71	57.99	Nucleus	795
*AoWRKY6*	9.44	31,856.07	290	50.58	59.83	Nucleus	873
*AoWRKY7*	5.41	31,262.65	285	49.21	56.18	Nucleus	858
*AoWRKY8*	6.04	46,486.38	413	73.69	73.20	Extracellular	1242
*AoWRKY9*	9.39	25,939.09	227	53.18	57.49	Nucleus	684
*AoWRKY10*	9.07	46,949.56	419	62.02	72.12	Nucleus	1260
*AoWRKY11*	8.97	13,655.35	122	33.04	51.89	Nucleus	369
*AoWRKY12*	7.67	20,371.41	177	56.61	49.49	Nucleus	534
*AoWRKY13*	5.61	27,669.79	244	51.91	63.52	Extracellular	735
*AoWRKY14*	8.38	21,604.16	199	55.32	55.88	Nucleus	600
*AoWRKY15*	6.21	63,743.09	585	49.54	56.02	Nucleus	1758
*AoWRKY16*	9.04	45,956.35	420	49.61	62.50	Nucleus	1263
*AoWRKY17*	8.53	44,932.82	399	57.93	58.87	Nucleus	1200
*AoWRKY18*	8.73	55,651.20	505	61.60	49.49	Nucleus	1518
*AoWRKY19*	8.81	21,958.55	196	60.57	51.79	Extracellular	591
*AoWRKY20*	5.65	30,925.36	272	50.22	57.79	Extracellular	819
*AoWRKY21*	9.54	20,022.42	174	40.10	58.16	Nucleus	525
*AoWRKY22*	6.92	40,959.89	360	52.21	59.36	Nucleus	1083
*AoWRKY23*	9.84	40,178.56	356	47.34	59.94	Nucleus	1071
*AoWRKY24*	7.67	38,127.89	345	57.86	52.87	Nucleus	1038
*AoWRKY25*	9.44	31,856.07	290	50.58	59.83	Nucleus	873
*AoWRKY26*	8.59	56,176.11	517	51.30	56.52	Nucleus	1554
*AoWRKY27*	5.12	28,238.23	253	52.89	61.66	Extracellular	762
*AoWRKY28*	8.77	24,308.45	209	48.08	60.05	Nucleus	630
*AoWRKY29*	4.66	19,682.85	181	49.78	51.22	Nucleus	546
*AoWRKY30*	5.18	34,275.83	300	76.79	65.97	Extracellular	903
*AoWRKY31*	8.34	26,410.43	230	58.47	61.43	Nucleus	693
*AoWRKY32*	5.57	72,655.57	674	61.01	53.25	Nucleus	2025
*AoWRKY33*	10.02	29,933.48	267	45.29	59.63	Nucleus	804
*AoWRKY34*	undefined	undefined	259	51.14	65.52	Nucleus	780
*AoWRKY35*	5.99	46,045.61	422	52.88	53.39	Nucleus	1269
*AoWRKY36*	10.1	17,298.73	156	56.66	52.63	Nucleus	471
*AoWRKY37*	5.54	32,599.17	298	59.24	49.43	Extracellular	897
*AoWRKY38*	10.24	24,637.92	222	58.85	65.45	Extracellular	669
*AoWRKY39*	5.06	31,499.98	278	58.6	73.31	Nucleus	837
*AoWRKY40*	4.89	32,499.36	293	41.65	66.93	Extracellular	882
*AoWRKY41*	8.56	25,121.05	219	43.79	55.25	Extracellular	660
*AoWRKY42*	5.54	32,599.17	298	59.24	49.43	Extracellular	897
*AoWRKY43*	5.22	35,686.71	327	60.99	68.01	Extracellular	984
*AoWRKY44*	6.74	54,215.42	487	54.77	49.47	Chloroplast	1464
*AoWRKY45*	5.36	34,350.08	313	54.95	59.87	Nucleus	942
*AoWRKY46*	9.62	47,107.05	428	61.23	59.77	Extracellular	1287
*AoWRKY47*	7.72	34,634.99	315	65.79	63.14	Nucleus	948
*AoWRKY48*	9.28	31,193.24	281	64.28	61	Nucleus	846
*AoWRKY49*	8.32	46,637.34	423	56.38	50.45	Nucleus	1272
*AoWRKY50*	6.31	30,633.89	277	70.53	46.14	Nucleus	834
*AoWRKY51*	9.9	36,191.16	318	43.8	60.13	Nucleus	957
*AoWRKY52*	7.78	12,996.32	114	42.92	39.3	Nucleus	345
*AoWRKY53*	5.49	32,548.79	299	43.69	52.51	Nucleus	900
*AoWRKY54*	9.17	19,282.63	169	51.94	62.31	Nucleus	510
*AoWRKY55*	7.05	22,006.13	200	53.76	56.55	Nucleus	603
*AoWRKY56*	6.2	57,636.71	529	52.79	54.97	Nucleus	1590
*AoWRKY57*	5.85	59,592.16	541	50.51	58.15	Nucleus	1626
*AoWRKY58*	9.38	14,978.79	136	76.11	53.82	Nucleus	411
*AoWRKY59*	6.82	14,042.8	122	70.52	58.36	Nucleus	369
*AoWRKY60*	6.55	27,418.73	242	44.53	64.88	Nucleus	729
*AoWRKY61*	6.11	25,787.13	232	60.7	75.26	Extracellular	699
*AoWRKY62*	4.83	32,010.46	283	57.63	60.99	Extracellular	852
*AoWRKY63*	6.35	27,294.78	241	52.99	68.84	Extracellular	726
*AoWRKY64*	6.76	30,279.76	266	60.72	61.62	Extracellular	801
*AoWRKY65*	5.88	31,046.72	277	57.25	78.19	Extracellular	834
*AoWRKY66*	6.37	26,946.25	238	54.49	61.89	Extracellular	717
*AoWRKY67*	5.29	40,200.29	356	43.22	78.99	Extracellular	1071
*AoWRKY68*	9.17	18,202.33	154	47.22	54.35	Nucleus	465
*AoWRKY69*	8.97	23,501.23	204	44.12	48.24	Nucleus	615
*AoWRKY70*	6.26	33,056.91	297	53.52	60.77	Extracellular	894

**Table 2 genes-14-01704-t002:** Tandem duplication of *AoWRKY* family and Ka, Ks values.

Tandem-Duplicated Gene Pairs	Ka	Ks	Ka/Ks	Chromosome	Subfamily
*AoWRKY47* & *AoWRKY48*	0.194258151	0.418275774	0.464426015	Chr7	C4
*AoWRKY64* & *AoWRKY65*	0.272881601	0.909314382	0.300095992	Chr8	C8

## Data Availability

Data are contained within the article or Appendix A.

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
