# Peer review of "Genome-Wide Identification and Analysis of the WRKY Gene Family in Asparagus officinalis"

_genes, 2023, doi:10.3390/genes14091704_

Round 1
Reviewer 1 Report
The title seems too complicated, please make it understandable.
Write the full name where the abbreviations are first used.
There are too many abbreviations in the abstract, more numerical data can be added to the abstract section.
line 45
Pay attention to the spelling of units. What is C2H2? use subscript.
line 67
Note the spelling of Latin. In Latin names, the second word is lowercase?
The introduction part of the study should be rewritten. Firstly, information about the plant studied should be given, and then trascripton information should be given.
In addition, information about the chromosome number of the plant should be given.
The full names of the abbreviations used in the tables should be written under the tables.
The spelling and names of herbal hormones should be checked.
The use of different contrasting colors in the phylogenetic tree may be more beneficial.
The conclusion part of the study should be a little more detailed.
Author Response
Dear reviewer, thank you very much for your comments. I have carefully studied and modified according to the requirements. The following is my modified content.
Point 1: The title seems too complicated, please make it understandable; Write the full name where the abbreviations are first used; There are too many abbreviations in the abstract, more numerical data can be added to the abstract section.
Response 1: Please provide your response for Point 1.
Dear reviewer, thank you for your comments. The following is my revision. I have simplified the topic and fully expressed the theme of this study; All the names that appear for the first time in the article and need to be explained have been written out in full, see the third paragraph of introduction for details; The summary has been carefully revised, and there are not too many acronyms.
Point 2: line 45,Pay attention to the spelling of units. What is C2H2? use subscript;line 67,Note the spelling of Latin. In Latin names, the second word is lowercase?The introduction part of the study should be rewritten. Firstly, information about the plant studied should be given, and then trascripton information should be given;In addition, information about the chromosome number of the plant should be given.
Response 2: Please provide your response for Point 2.
line 45 ,C2H2 has been changed to C2H2; line 67, Arabidopsis Location The first letter of the second word has been changed to lower case; The transcriptional information, whole genome information and chromosome information of asparagus have been added in the introduction, see the fourth paragraph of the introduction for details.
Point 3: The full name of the abbreviation used in the form should be written below the form;The spelling and names of herbal hormones should be checked;It may be more beneficial to use different contrasting colors in phylogenetic trees.The conclusion of the study should be more detailed.
Response 3: Please provide your response for Point 3.
The full names of the abbreviations in the table have been reflected in the paper, see Results 3.1 for details; Hormone names to add full names, see Results 3.3; There are many colors in the phylogenetic tree, so in addition to distinguishing different subgroups, the current colors are selected for the beauty and harmony of the overall picture. The conclusion of the study has been duly refined.
Reviewer 2 Report
Comments to the Authors
The manuscript entitled “Genome-Wide Identification and Analysis of the WRKY Transcription Factor Gene Family in Asparagus officinalis” extensively presented the WRKY transcription factor gene in Asparagus officinalis. The authors have presented the results most appropriately in figures and tables format. However, the following concerns should be addressed for the improvement of the manuscript.
General comments
The manuscript needs thorough revision for grammatical and formatting errors.
The scientific and gene names should be italicized.
Abstract
Line 16-17 needs to be revised to convey a comprehensive message.
The practical utility of the work needs to be mentioned at the end of the abstract.
Introduction
The introduction section is written more generally and needs to be revised extensively with a proper flow of content.
The authors emphasized solely on WRKY transcription factor. However, it needs to be presented in connection to the target crop Asparagus rather than the transcription factor mostly.
The area and production of Asparagus should be provided. The extent of yield loss due to various stresses must be mentioned.
Materials and methods
As all the data has been accessed one year before, it needs to be checked for any updates available to the presented data.
Results and Discussion
Figures 7A & B should be replaced with high-quality images and the abbreviation in the image needs to be explained in the figure legend.
The authors presented their results from the study, however, the discussion of the results in comparison with other studies needs to be improved.
Minor editing in the English language is required.
Author Response
Dear reviewer, thank you very much for your comments. I have carefully studied and modified according to the requirements. The following is my modified content.
Point 1: General comments: The manuscript needs thorough revision for grammatical and formatting errors.The scientific and gene names should be italicized.
Response 1: Please provide your response for Point 1.
The grammatical errors in the manuscript have been corrected; Gene names and Latin scientific names have all been italicized.
Point 2: Abstract:Line 16-17 needs to be revised to convey a comprehensive message.
The practical utility of the work needs to be mentioned at the end of the abstract.
Response 2: Please provide your response for Point 2.
Lines 16-17 have been revised to provide more comprehensive information. New content has been added at the end of the summary to make it more detailed.
Point 3: Introduction:The introduction section is written more generally and needs to be revised extensively with a proper flow of contentThe authors emphasized solely on WRKY transcription factor. However, it needs to be presented in connection to the target crop Asparagus rather than the transcription factor mostly.The area and production of Asparagus should be provided. The extent of yield loss due to various stresses must be mentioned.
Response 3: Please provide your response for Point 3.
Most of the content has been revised in the introduction. In addition to WRKY transcription factors, we have added detailed information of the target crop asparagus, including chromosome number, transcriptome information, yield and area information, etc.
Point 4: Materials and methods:As all the data has been accessed one year before, it needs to be checked for any updates available to the presented data.
Response 4: Please provide your response for Point 4.
For the data I accessed a year ago, I have rechecked the updates and there is no difference from a year ago.
Point 5: Results and Discussion:Figures 7A & B should be replaced with high-quality images and the abbreviation in the image needs to be explained in the figure legend.
The authors presented their results from the study, however, the discussion of the results in comparison with other studies needs to be improved.
Response 5: Please provide your response for Point 5.
Figures 7A and 7B have been replaced with higher quality images, and the abbreviations in the images have been added to the bottom of the image. Some results are added to the discussion section.
Reviewer 3 Report
The abstract is written very general type. Kindly improve and add few result also.
Kindly add the latest reference in the Introduction section of the manuscript.
Introduction part is very short.
In the methods section only weblink is enough no need accessed on 23 August 2022
Figure 2 and 6: Describe the legend section
I am not happy with the discussion section kindly improve and more descriptive way.
Check the grammatical mistake throughout the manuscript
Minor editing of English language required
Author Response
Dear reviewer, thank you very much for your comments. I have carefully studied and modified according to the requirements. The following is my modified content.
Point 1: The abstract is written very general type. Kindly improve and add few result also.Kindly add the latest reference in the Introduction section of the manuscript.
Introduction part is very short.In the methods section only weblink is enough no need accessed on 23 August 2022.Figure 2 and 6: Describe the legend section I am not happy with the discussion section kindly improve and more descriptive way.Check the grammatical mistake throughout the manuscript.
Response 1: Please provide your response for Point 1.
The summary section has been revised for the most part, adding more detailed results at the end. The introduction has been substantially revised, with the addition of part of the 2022 literature and the latest asparagus yield information for 2023. We have increased the length of the introduction appropriately. The methods section keeps only the web address. Figure 2 and Figure 6 add the legend section as appropriate. The discussion part has been added in a larger space. The grammatical errors in the article have been corrected.
Reviewer 4 Report
In this MS, Chen et al. specifically focus on the WRKY gene family in Asparagus officinalis. To gain a broader understanding of the WRKY gene family, the authors performed comprehensive studies including the identification of WRKY gene family members, construction of a phylogenetic tree, examination of conserved motifs and gene domains, extraction of cis-acting elements, analysis of intron/exon distribution, species collinearity assessment, and exploration of WRKY expression patterns under salt stress conditions. In essence, this study aims to leverage the extensive analytical insights into the WRKY family to generate novel ideas for future research endeavors pertaining to this gene family's intricate functions.
Overall speaking, the study is good and fits into the scope of “Genes”; and has interest not only in gene family evolution but also in the study of Asparagus officinalis. However, this reviewer still has some questions. For details, see below:
1. The authors constructed one gene tree and one protein tree in the MS. This reviewer would like to know what outgroup did the authors use to construct the trees. Different outgroup may result in different phylogenetic pattern. Please clarify the reason for choosing the outgroup or reason not use the outgroup.
2. It seems the line 197 to 200 needs citations.
3. Line 211, are you going to say “one gene” instead of “one chromosome”?
4. Line 238, “Structure” should be “structure”.
5. Line 315 to 317, this reviewer is confused with the criteria to define decreased genes and increased genes.
6. For Figure 7A/B, the reproducibility of RNAseq is not good. For some genes, the expression patterns are different among the three biological replicates. How can we trust these data?
7. AoWRKY30 is increased in Figure 7C, but this gene is decreased in Figure 7A. The pattern for AoWRKY11 and AoWRKY12 are also contradictory. So did you compare NIS to NI or NI to NIS? Please clarify this part.
8. The gene expression pattern seems quite different between Figure 7A and 7B. How to explain this difference? Which dataset should we trust and why?
Author Response
Dear reviewer, thank you very much for your comments. I have carefully studied and modified according to the requirements. The following is my modified content.
Point 1: The authors constructed one gene tree and one protein tree in the MS. This reviewer would like to know what outgroup did the authors use to construct the trees. Different outgroup may result in different phylogenetic pattern. Please clarify the reason for choosing the outgroup or reason not use the outgroup.
Response 1: Please provide your response for Point 1.
When constructing the evolutionary tree, we chose Arabidopsis Thaliana as a group with evolutionary relationship, mainly because Arabidopsis thaliana as a model plant has been studied more thoroughly, and the WRKY gene family of Arabidopsis Thaliana has been reported by many researchers, and the function of the gene is relatively complete, which can provide us with a good reference. Appropriate explanations are added in the paper, which are reflected in Result 3.2.
Point 2: It seems the line 197 to 200 needs citations.
Response 2: Please provide your response for Point 2.
lines 197 to 200 are from the website TAIR and have been identified in the article.
Point 3: Line 211, are you going to say “one gene” instead of “one chromosome”?
Response 3: Please provide your response for Point 3.
"Gene" has been changed to "chromosome".
Point 4: Line 238, “Structure” should be “structure”.
Response 4: Please provide your response for Point 4.
"Structure" has been changed to "structure".
Point 5: Line 315 to 317, this reviewer is confused with the criteria to define decreased genes and increased genes.
Response 5: Please provide your response for Point 5.
The criteria for gene down-regulation and up-regulation have been added to the corresponding references.
Point 6: For Figure 7A/B, the reproducibility of RNAseq is not good. For some genes, the expression patterns are different among the three biological replicates. How can we trust these data?
Response 6: Please provide your response for Point 6.
The data reproducibility of some genes in FIG. 7A and FIG. 7B was poor. We guessed that some technical errors occurred during the experiment, which led to unsatisfactory data of some samples of these genes, such as the expression level of some genes being 0. So we decided to focus only on genes that were significantly up-regulated and down-regulated.
Point 7: AoWRKY30 is increased in Figure 7C, but this gene is decreased in Figure 7A. The pattern for AoWRKY11 and AoWRKY12 are also contradictory. So did you compare NIS to NI or NI to NIS? Please clarify this part.
Response 7: Please provide your response for Point 7.
There was no clear standard for the result of this part before, and I have elaborated on it in detail in the result part. The expression pattern in FIG. 7A and FIG. 7C is the same. The reason for the confusion is that we did not give reference objects before.
Point 8: The gene expression pattern seems quite different between Figure 7A and 7B. How to explain this difference? Which dataset should we trust and why?
Response8: Please provide your response for Point 8.
FIG. 7A and FIG. 7B are different treatment criteria. FIG. 7A mainly shows the effect of salt stress on gene expression, which is the result of regulation by a single variable. In addition to the effects of salt stress, salt-tolerant strains were also added in FIG. 7B. In this study, we mainly analyzed expression patterns in individual groups, but we also added some discussion about four groups of treatment to the discussion
Round 2
Reviewer 1 Report
ms is ready for publication
Reviewer 3 Report
The authors has revised the Manuscript very well as per the reviewers comments. My decision is accepted the MS for publication.
Minor editing of English language required